# Epigallocatechin-3-Gallate Prevents Acute Gout by Suppressing NLRP3 Inflammasome Activation and Mitochondrial DNA Synthesis

**DOI:** 10.3390/molecules24112138

**Published:** 2019-06-06

**Authors:** Hye Eun Lee, Gabsik Yang, Youn Bum Park, Han Chang Kang, Yong-Yeon Cho, Hye Suk Lee, Joo Young Lee

**Affiliations:** BK21 Plus, College of Pharmacy, The Catholic University of Korea, Bucheon 14662, Korea; esthel0513@catholic.ac.kr (H.E.L.); yangboncho@gmail.com (G.Y.); pyb12345678@naver.com (Y.B.P.); hckang@catholic.ac.kr (H.C.K.); yongyeon@catholic.ac.kr (Y.-Y.C.); sianalee@catholic.ac.kr (H.S.L.)

**Keywords:** inflammasome, gout, mitochondria, reactive oxygen species, green tea, macrophages, innate immunity

## Abstract

Gout is a chronic inflammatory disease evoked by the deposition of monosodium urate (MSU) crystals in joint tissues. The nucleotide-binding oligomerization domain (NOD)-like receptor (NLR) family pyrin domain containing 3 (NLRP3) inflammasome is responsible for the gout inflammatory symptoms induced by MSU crystals. We investigated whether epigallocatechin-3-gallate (EGCG) suppresses the activation of the NLRP3 inflammasome, thereby effectively preventing gouty inflammation. EGCG blocked MSU crystal-induced production of caspase-1(p10) and interleukin-1β in primary mouse macrophages, indicating its suppressive effect on the NLRP3 inflammasome. In an acute gout mouse model, oral administration of EGCG to mice effectively alleviated gout inflammatory symptoms in mouse foot tissue injected with MSU crystals. The *in vivo* suppressive effects of EGCG correlated well with the suppression of the NLRP3 inflammasome in mouse foot tissue. EGCG inhibited the *de novo* synthesis of mitochondrial DNA as well as the production of reactive oxygen species in primary mouse macrophages, contributing to the suppression of the NLRP3 inflammasome. These results show that EGCG suppresses the activation of the NLRP3 inflammasome in macrophages via the blockade of mitochondrial DNA synthesis, contributing to the prevention of gouty inflammation. The inhibitory effects of EGCG on the NLRP3 inflammasome make EGCG a promising therapeutic option for NLRP3-dependent diseases such as gout.

## 1. Introduction

Gout is a relatively common inflammatory arthritis that can cause pain in the joints and seriously impair the quality of life of a patient. Gout is typically known to occur in middle-aged men; yet, the incidence is increasing in the elderly population [1]. The precipitation of monosodium urate (MSU) crystals in joints is a key factor in the initiation and development of gout [2].

Accumulating evidence indicates that the MSU crystals-induced inflammatory responses and gout pathogenesis are dependent on interleukin (IL)-1β [3]. It is now well established that the nucleotide-binding oligomerization domain (NOD)-like receptor (NLR) family pyrin domain containing 3 (NLRP3) inflammasome senses MSU crystal deposition, subsequently activating the downstream immune signals and inducing the production of IL-1β [3]. The NLRP3 inflammasome is a multiprotein complex consisting of NLRP3, apoptosis-associated speck-like protein containing a CARD (ASC) as an adaptor, and pro-caspase-1 as an effector enzyme [4]. Activation of the NLRP3 inflammasome culminates in the activation of pro-caspase-1, leading to the cleavage of inactive pro-IL-1β and pro-IL-18 to the active form of IL-1β and IL-18. Mice deficient in NLRP3 fail to produce active IL-1β in the foot pads in response to MSU crystals in an acute gout mouse model [5].

Current therapeutic strategies to target IL-1β have proven successful for alleviating the symptoms of gout in clinical studies, suggesting that targeting the NLRP3 inflammasome may have a critical impact on gout treatment. Current clinical trials are underway for the development of IL-1 inhibitors including anakinra (an IL-1 receptor antagonist) [6], rilonacept (IL-1 Trap, a soluble decoy receptor) [7], and canakinumab (anti-IL-1β monoclonal antibody) [8]. Despite the efficacy of these inhibitors, there are some issues in patients, limiting the extensive use of IL-1 inhibitors in the treatment of gout. These include the high cost and the inconvenient administration route of IL-1 inhibitors [9].

Therefore, we searched for orally available small-molecule drugs for gout treatment that inhibit the NLRP3 inflammasome, thereby blocking the production of active IL-1β. Epigallocatechin-3-gallate (EGCG) is an active component of green tea and is well known to have anti-inflammatory properties. There is a previous report that EGCG reduces the messenger RNA (mRNA) and protein expression of renal NLRP3 in a lupus nephritis mouse model [10]. Similarly, Gao *et al*. reported that EGCG decreases the protein level of renal NLRP3 in a contrast-induced nephropathy rat model [11]. These reports show that oral or intravenous infusion of EGCG suppresses the expression of tissue NLRP3 protein, possibly leading to the reduction of NLRP3 inflammasome activation. However, it is not entirely clear whether EGCG affects the activation of NLRP3 inflammasome induced by various NLRP3 inducers in macrophages and in gouty inflammatory response. In this study, we investigated whether EGCG could suppress the activation of the NLRP3 inflammasome in macrophages in order to pursue the possibility of its therapeutic application for NLRP3-mediated diseases such as gout.

## 2. Results

### 2.1. EGCG Suppresses NLRP3 Inflammasome Activation in Primary Macrophages

We investigated whether EGCG blocks the activation of the NLRP3 inflammasome induced by various activators in primary mouse macrophages. To exclude the possibility that EGCG may affect the priming activation step by lipopolysaccharide (LPS), macrophages were treated with EGCG after the LPS was washed out. Cleavage of pro-caspase-1 and pro-IL-1β to caspase-1(p10) and IL-1β, respectively, is considered a hallmark of inflammasome activation. EGCG inhibited the MSU crystal-induced cleavage of pro-caspase-1 and pro-IL-1β to caspase-1(p10) and IL-1β in cell supernatants, as shown by immunoblotting (Figure 1A). Consistently, EGCG reduced the MSU crystal-induced secretion of IL-1β in a dose-dependent manner, as measured by enzyme-linked immunosorbent assay (ELISA) of the cell culture medium (Figure 1B).

We further examined whether EGCG inhibits NLRP3 inflammasome activation by other activators such as adenosine triphosphate (ATP) and nigericin. EGCG suppressed ATP-induced cleavage of pro-caspase-1 and pro-IL-1β to caspase-1(p10) and IL-1β in primary macrophages (Figure 1C). In addition, ATP-induced IL-1β secretion was suppressed by EGCG (Figure 1D). Nigericin-induced the cleavage of pro-caspase-1 and pro-IL-1β to caspase-1(p10) and IL-1β was blocked by EGCG (Figure 1E). The nigericin-induced secretion of IL-1β was reduced by EGCG (Figure 1F).

These results show that EGCG suppresses NLRP3 inflammasome activation induced by MSU crystals, ATP, and nigericin, suggesting that EGCG inhibits NLRP3 inflammasome activation by various stimuli.

### 2.2. Oral Administration of EGCG Prevents Acute Gouty Inflammation in Mice by Blocking NLRP3 Inflammasome Activation

We investigated whether the inhibitory effects of EGCG on MSU crystal-induced NLRP3 inflammasome activation resulted in the prevention of gouty inflammation in an acute gout mouse model. An acute gout mouse model was generated by injecting MSU crystals into the hindfoot; the injection led to increased footpad thickness (Figure 2A). In contrast, oral administration of EGCG reduced the footpad thickness to normal levels in a dose-dependent manner (Figure 2A). EGCG blocked the MSU crystal-induced recruitment of neutrophils to foot tissues as shown by histological examination (Figure 2B). In addition, EGCG suppressed the recruitment of myeloperoxidase to the MSU crystal-injected foot tissues as determined by immunohistochemical staining (Figure 2C). Furthermore, EGCG prevented the MSU crystal-induced production of inflammatory cytokines such as IL-6 in the foot tissues (Figure 2D). These results show that oral administration of EGCG attenuates the inflammatory symptoms of acute gout caused by the injection of uric acid crystals in mice.

We examined whether EGCG suppresses NLRP3 inflammasome activation in foot tissues injected with MSU crystals. Injection of MSU crystals induced the cleavage of pro-caspase-1 to caspase-1(p10) and the cleavage of pro-IL-1β to IL-1β in the foot tissue homogenates (Figure 3A). The oral administration of EGCG prevented the cleavage of pro-caspase-1 to caspase-1(p10) and of pro-IL-1β to IL-1β in the foot tissues injected with MSU crystals (Figure 3A). EGCG decreased MSU crystal-induced IL-1β production in foot tissue homogenates as determined by ELISA (Figure 3B). Immunohistochemical analysis of the mouse foot tissues with anti-caspase-1 and IL-1β antibody staining showed that the levels of caspase-1 and IL-1β were increased in the foot tissues injected with the MSU crystals, while oral administration of EGCG reduced the increased levels of caspase-1 and IL-1β in gouty foot tissues (Figure 3C).

These results demonstrate that oral administration of EGCG effectively alleviates the inflammatory symptoms of uric acid crystal-induced acute gout in mice, mediated by blockade of NLRP3 inflammasome activation.

### 2.3. EGCG Inhibits the De Novo Synthesis of Mitochondrial DNA in Macrophages

To investigate the mechanism by which EGCG suppresses NLRP3 inflammasome activation, we investigated whether EGCG directly binds to the components of the NLRP3 inflammasome by surface plasmon resonance (SPR) analysis with a recombinant protein, the PYD of NLRP3 or ASC. The SPR analysis demonstrated that EGCG did not directly associate with the PYD of NLRP3 or ASC (Figure 4A,B). We further investigated whether EGCG inhibits the enzymatic activity of caspase-1 by *in vitro* caspase-1 enzyme activity using recombinant caspase-1. EGCG did not suppress the enzymatic activity of caspase-1, while a caspase inhibitor, Z-YVAD-FMK, decreased the caspase-1 enzymatic activity (Figure 4C). These results show that EGCG does not directly inhibit NLRP3 inflammasome components such as NLRP3, ASC, and caspase-1.

Activation of the NLRP3 inflammasome causes mitochondrial disruption, leading to the generation of reactive oxygen species (ROS) [12]. Therefore, we investigated whether EGCG reduces the generation of ROS induced by NLRP3 inflammasome activation in primary mouse macrophages. EGCG suppressed intracellular ROS production induced by the MSU crystals in macrophages (Figure 5A). Consistently, ROS production induced by ATP or nigericin was significantly decreased by EGCG (Figure 5B,C).

ROS generated upon NLRP3 inflammasome activation induce the oxidation of mitochondrial DNA (mtDNA), and oxidized mtDNA is released into the cytosol to induce the activation of the NLRP3 inflammasome [13]. It has been reported that mtDNA is synthesized *de novo* during the process of NLRP3 inflammasome activation [13]. Therefore, we investigated whether EGCG affects the *de novo* synthesis of mtDNA in macrophages. Relative total mtDNA amounts were quantified by quantitative PCR with primers specific for the mitochondrial D-loop region (D-loop) or a region of mtDNA that is not inserted into nuclear DNA (non-NUMT) in macrophages. The amount of nuclear DNAs (Tert, B2m) was measured to normalize mtDNA production. MSU crystals increased the amount of mtDNA (D-loop, non-NUMT), while EGCG suppressed MSU crystal-induced mtDNA synthesis (Figure 5D,E). These results show that EGCG blocks the *de novo* synthesis of mtDNA occurring upon NLRP3 activation in macrophages. These results suggest that EGCG reduces the production of oxidized mtDNA by suppressing both ROS production and mtDNA synthesis.

Collectively, these results show that EGCG suppresses the activation of the NLRP3 inflammasome in macrophages via the blockade of mtDNA synthesis, contributing to the prevention of gouty inflammation.

## 3. Discussion

EGCG, a green tea polyphenol component, is well known for its anti-inflammatory properties. Previous reports have shown that the inhibition of the NLRP3 inflammasome by EGCG may lead to the suppression of lupus nephritis and peritonitis in mouse models [14,15]. Our results further demonstrate that EGCG is effective in preventing acute gout by inhibiting NLRP3 inflammasome activation, as oral administration of EGCG reduced inflammatory symptoms in an MSU crystal-induced acute gout mouse model. The inhibitory effects of EGCG on gouty inflammation correlate well with decreased NLRP3 inflammasome activity in gouty tissues. EGCG treatment reduced IL-1β secretion and neutrophil recruitment in foot tissues injected with MSU crystals. Gout is a form of inflammatory arthritis that is induced by the deposition of uric acid crystals, characterized by neutrophil infiltration into the inflammatory joints. Discovering the role of the NLRP3 inflammasome and the subsequent release of IL-1β is important to elucidate the pathogenesis of this disease [16]. Our study suggests that administration of EGCG could be extended to the treatment of other diseases relevant to uric acid crystal accumulation and the NLRP3 inflammasome.

MSU crystals, ATP, and nigericin have different upstream signaling pathways to activate NLRP3. MSU crystals are phagocytosed and destabilize the phagosome, which activates the NLRP3 inflammasome. ATP triggers NLRP3 inflammasome activation by binding to the P2X7 purinergic receptor, thereby increasing potassium efflux. Nigericin, a microbial toxin derived from *Streptomyces hygroscopicus*, decreases intracellular potassium levels by acting as a potassium ionophore. Our results show that in addition to suppressing MSU-induced NLRP3 activation, EGCG treatment suppresses various activators of the NLRP3 inflammasome, such as ATP and nigericin, suggesting that the inhibitory target of EGCG may be common to these different activators. We previously reported that the inhibitory effects of caffeic acid phenethyl ester are due to the direct binding of caffeic acid phenethyl ester to ASC [5]. In contrast, the SPR analysis showed that EGCG does not directly bind to ASC or NLRP3 PYD. EGCG also does not inhibit caspase-1 enzymatic activity. These results demonstrate that the target of EGCG does not lie in the NLRP3 inflammasome complex itself but in the upstream pathways.

Oxidized mtDNA is known to activate the NLRP3 inflammasome [17]. Zhong *et al*. showed that NLRP3 priming induces *de novo* synthesis of mtDNA, which is further oxidized under oxidative stress conditions [13]. Cytidine/uridine monophosphate kinase 2 (CMPK2) is an essential enzyme that provides deoxyribonucleotides for mtDNA synthesis. CMPK2 expression is regulated by interferon regulatory factor 1 (IRF1) transcription factor activated in an LPS-priming step [13]. CMPK2-dependent mtDNA synthesis is required for the production of oxidized mtDNA fragments in response to NLRP3 activators [13]. Oxidized mtDNA associates with the NLRP3 inflammasome complex to activate NLRP3 inflammasome. We show that *de novo* synthesis of mtDNAs induced by MSU crystals is decreased by EGCG in macrophages. This suggests that EGCG may regulate the upstream events to produce mtDNA including CMPK2 activity or CMPK2 expression. The detailed mechanism for inhibition of *de novo* synthesis of mtDNA by EGCG needs to be investigated further. Blockade of mtDNA synthesis inhibited IL-1β production induced by NLRP3 activators such as ATP, nigericin, and MSU, while IL-1β production by an AIM2 inflammasome activator, poly dA:dT, was not inhibited by blockade of mtDNA synthesis [13]. These show that newly synthesized mtDNA is required for NLRP3 inflammasome activation induced by various NLRP3 activators [13]. Therefore, it is likely that the common inhibitory effect of EGCG on various NLRP3 inducers may be linked to the common suppression of mtDNA synthesis in response to various NLRP3 inducers including ATP or nigericin. This needs to be further elucidated in a future study. In addition, EGCG reduces the ROS levels increased by MSU crystal stimulation in macrophages. These results suggest that the suppression of NLRP3 inflammasome activation by EGCG is mediated by the blockade of mtDNA synthesis, which is the source of oxidized mtDNA, an NLRP3 activator. EGCG appears to be effective in reducing the generation of oxidized mtDNAs by blocking both new mtDNA synthesis and ROS production.

In conclusion, EGCG inhibits acute gout inflammation, including proinflammatory cytokine release and neutrophil infiltration into the lesion site, mediated by the suppression of NLRP3 inflammasome activation in macrophages. The inhibition of ROS production and mtDNA synthesis by EGCG contributes to its suppressive effect on the NLRP3 inflammasome.

## 4. Materials and Methods

### 4.1. Animals and Cell Culture

Mice (C57BL/6) were obtained from RaOn Bio (Seoul, Korea). The mice were housed in a room controlled for temperature (23 ± 3 °C) and relative humidity (40–60%) under specific pathogen-free conditions. Mice were acclimated in the animal facility for at least a week before the experiments. Animal care and the experimental protocols were carried out in accordance with the guidelines of the Institutional Animal Care and Use Committee (IACUC) of the Catholic University of Korea (permission #2014-015). Bone marrow-derived primary macrophages (BMDMs) were prepared after bone marrow was isolated from C57BL/6 mice as described previously [18]. Cell culture was performed as previously described [19]. Briefly, BMDMs were cultured in Dulbecco’s modified eagle medium (DMEM) containing 10% (*v*/*v*) fetal bovine serum (Invitrogen, Carlsbad, CA, USA), 10,000 units/mL of penicillin, and 10,000 μg/mL of streptomycin (Invitrogen).

### 4.2. Reagents

Epigallocatechin-3-gallate (EGCG) was purchased from Sigma-Aldrich (St. Louis, MO, USA) and the stock solution was prepared in DMSO. Purified LPS from *Escherichia coli* was obtained from List Biological Laboratory Inc. (Campbell, CA, USA) and dissolved in endotoxin-free water. Monosodium urate (MSU) and ATP were purchased from Invivogen (Carlsbad, CA, USA). Nigericin and H_2_DCFDA were purchased from Sigma-Aldrich. A caspase inhibitor, Z-YVAD-FMK, was obtained from Calbiochem (Darmstadt, Germany). An antibody for mouse caspase-1 was obtained from Santa Cruz Biotechnology (Santa Cruz, CA, USA). An antibody for interleukin-1β (IL-1β) was from R&D Systems (Minneapolis, MN, USA). All other reagents, if not specified, were purchased from Sigma-Aldrich.

### 4.3. Analysis of Inflammasome Activation

This was performed as previously described [20]. BMDMs were plated in 6-well plates at a density of 2 × 10^6^ cells/mL and primed with LPS (500 ng/mL for MSU crystals stimulation; 100 ng/mL for ATP, and nigericin stimulation) for 4 h. To exclude the effect of EGCG on the LPS priming step, EGCG was added after washing out the LPS with phosphate-buffered saline (PBS). The cells were treated with EGCG for 1 h and stimulated with NLRP3 inflammasome activators such as MSU crystals, ATP, and nigericin in the presence or absence of EGCG in serum-free medium. The cells were lysed in radioimmunoprecipitation assay (RIPA) buffer (50 mM Tris-HCl, pH 7.4, 1% NP-40, 0.25% sodium deoxycholate, 150 mM NaCl, 1 mM EGTA, 1 mM PMSF, 1 mM Na_3_VO_4_, 10 μg/mL aprotinin, and 10 μg/mL leupeptin). The degradation of pro-caspase-1 to caspase-1(p10) and the cleavage of pro-IL-1β to IL-1β in the supernatants were regarded as indicators of inflammasome activation and were determined by immunoblot assays.

### 4.4. Enzyme-Linked Immunosorbent Assays

Enzyme-linked immunosorbent assays (ELISAs) were performed as previously described [21]. Levels of IL-1β and IL-6 in culture media or foot pad homogenates were determined using DuoSet enzyme-linked immunosorbent assay (ELISA) kit (R&D Systems, Minneapolis, MN, USA).

### 4.5. An Acute Gout Mouse Model

An acute gout mouse model was performed as previously described [5]. Briefly, C57BL/6 mice (7 to 8 weeks old) were orally administered with 0.5 mL sterilized water containing EGCG (1, 10, 30 mg/kg) or vehicle (0.02% DMSO). After 1 h, MSU crystals (2 mg in 0.1 mL of sterile, endotoxin-free PBS) or PBS were subcutaneously injected under the plantar surface of the right paw. The foot thickness was monitored over time. Twenty-four hours after injecting the MSU crystals, the foot tissues were homogenized in RIPA buffer, and the supernatant was collected for ELISAs and immunoblot assays. For histological analysis, sagittal sections of the footpads were fixed in 10% paraformaldehyde and stained with hematoxylin and eosin. For immunohistochemistry, the sections were deparaffinized in xylene and then rehydrated using alcohol series. Primary antibody was diluted 1:200. The sections were then incubated with secondary antibody at room temperature for 30 min.

### 4.6. Caspase-1 Enzyme Activity Assay

The enzymatic activity of caspase-1 was determined using a Caspase-1 assay kit from Bio-vision (Milpitas, CA, USA) according to the manufacturer’s instructions. Fluorescence was recorded at 400 nm after excitation at 505 nm using a SpectraMaxM5 microplate reader (Molecular Devices, Sunnyvale, CA, USA).

### 4.7. Reactive Oxygen Species (ROS) Measurement

ROS measurement was performed using H_2_DCFDA according to the manufacturer’s instructions. In brief, bone marrow-derived primary mouse cells were seeded in 96-well plates and incubated overnight. Macrophages were pre-treated with an oxidative stress indicator, H_2_DCFDA (10 μM) for 1 h and further stimulated with monosodium uric acid (MSU) crystals (500 μg/mL), ATP (5 mM), or nigericin (10 μM) for 1 h in the presence or absence of EGCG. Cells were washed with PBS and fluorescence intensity (excitation = 485 nm; emission = 530 nm) were measured using a microtiter plate reader (VICTOR^TM^X3, PerkinElmer, MA, USA).

### 4.8. Quantitative Real-Time Polymerase Chain Reaction (qPCR) Analysis for Total Mitochondrial DNA

Total mitochondrial DNAs (mtDNAs) were isolated using G-spin^TM^ DNA Mini Kit (iNtRON Biotechnology, Korea) according to manufacturer’s instructions. qPCR was performed as previously described [22]. mtDNAs were quantified by qPCR using primers specific for the mitochondrial D-loop region (D-loop) or a specific region of mtDNA that is not inserted into nuclear DNA (non-NUMT). Nuclear DNA encoding Tert or B2m was used for normalization. Primer sequences are as follows: D-loop F: 5′-AATCTACCATCCTCCGTGAAACC-3′; D-loop R: 5′-TCAGTTTAGCTACCCCCAAGTTTAA-3′; Tert F: 5′-CTAGCTCATGTGTCAAGACCCTCTT-3′; Tert R: 5′-GCCAGCACGTTTCTCTCGTT-3′; B2m F: 5′-ATGGGAAGCCGAACATACTG-3′; B2m R: 5′-CAGTCTCAGTGGGGGTGAAT-3′; non-NUMT F: 5′-CTAGAAACCCCGAAACCAAA-3′; non-NUMT R: 5′-CCAGCTATCACCAAGCTCGT-3′. Specificity of the amplified PCR products was assessed by melting curve analysis.

### 4.9. Statistical Analysis

Data are expressed as means ± SD. Comparisons of data between groups were performed by one-way ANOVA followed by Tukey’s multiple range test. Values of *p* < 0.05 were considered significant. Representative data are presented from two or three independent experiments.

## Figures and Tables

**Figure 1 molecules-24-02138-f001:**
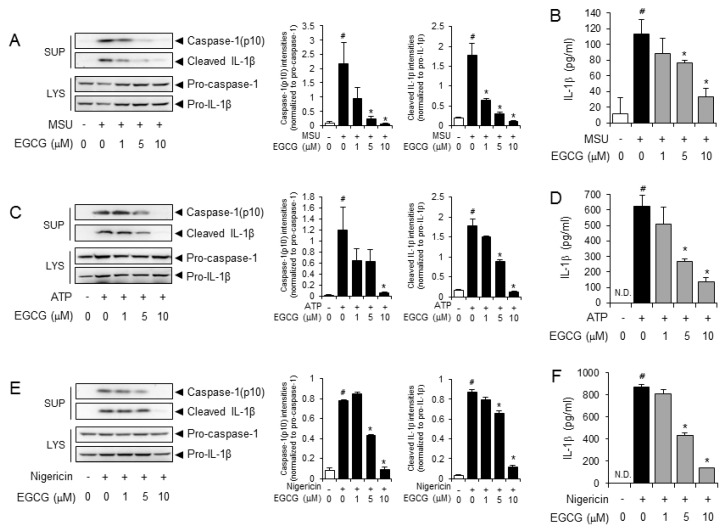
Epigallocatechin-3-gallate (EGCG) suppresses the activation of the (NOD)-like receptor (NLR) family pyrin domain containing 3 (NLRP3) inflammasome in primary macrophages. Lipopolysaccharide (LPS)-primed bone marrow-derived primary macrophages were treated with EGCG for 1 h and then stimulated with monosodium uric acid (MSU) crystals (500 μg/mL for 6 h), adenosine triphosphate (ATP) (5 mM for 1 h), or nigericin (10 μM; 2 h for E and 16 h for F) in the presence or absence of EGCG as indicated. (**A**,**C**,**E**) Cell culture supernatants (SUP) and cell lysates (LYS) were immunoblotted for pro-caspase-1, caspase-1(p10), pro-IL-1β, and IL-1β. In the bar graph, the band density of caspase-1(p10) and cleaved IL-1β normalized to corresponding pro-caspase-1 and pro-IL-1β, was expressed as means ± SD (*n* = 3). *#*, significantly different from vehicle group, *p* < 0.05; *, significantly different from MSU, ATP, or nigericin alone, *p* < 0.05. (**B**,**D**,**F**) Cell culture supernatants were analyzed for secreted IL-1β by ELISA. The values represent the means ± SD (*n* = 3). *#*, significantly different from vehicle group, *p* < 0.05; *, significantly different from MSU, ATP, or nigericin alone, *p* < 0.05. N.D., not detected. IL: interleukin.

**Figure 2 molecules-24-02138-f002:**
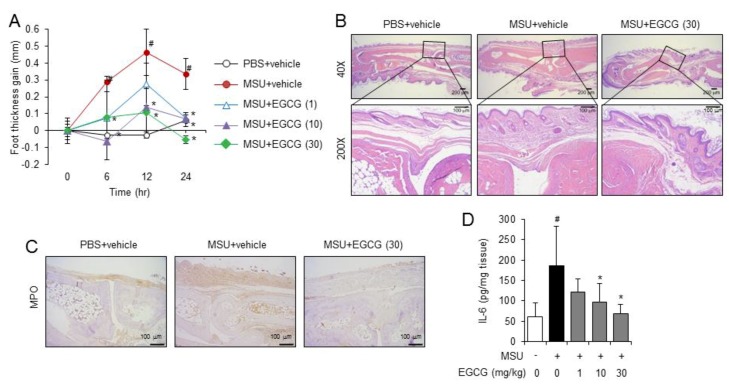
Oral administration of EGCG suppresses acute gout symptoms induced by MSU crystal injection in mice. Mice were orally administered EGCG (1, 10, and 30 mg/kg) or vehicle (0.02% dimethyl sulfoxide (DMSO) in water). After 1 h, MSU crystals (2 mg/0.1 mL of phosphate-buffered saline (PBS)/mouse) or PBS was subcutaneously injected into the footpad of the right hindfoot of each mouse. After 24 h, the footpad tissues were collected for further analysis. (**A**) Time course of footpad thickness gain compared with the footpad thickness at the 0 h time point per group. (**B**) Representative picture of hematoxylin and eosin staining of the hind feet. Infiltrating neutrophils in the hindfoot tissue appear as purple dots. (**C**) Representative pictures of immunohistochemistry staining of foot tissues with myeloperoxidase (MPO) (200×). (**D**) Supernatants of the foot tissue homogenates were analyzed for IL-6 by ELISA. The values in the line and bar graphs represent the means ± SD (*n* = 6 mice/group). *#*, significantly different from vehicle group, *p* < 0.05; *, significantly different from MSU alone, *p* < 0.05.

**Figure 3 molecules-24-02138-f003:**
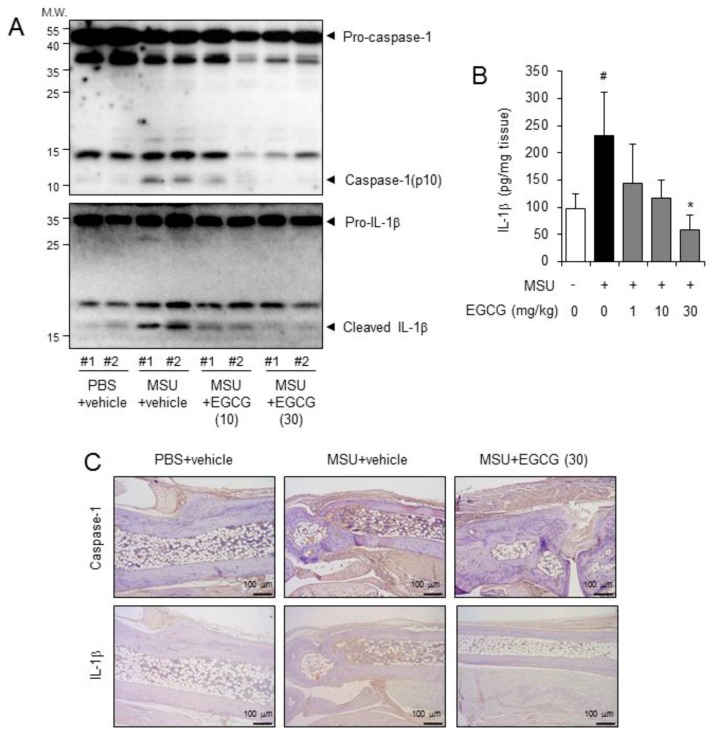
Oral administration of EGCG attenuates NLRP3 inflammasome activation in an acute gout mouse model. The foot tissue samples analyzed are the same as those used in Figure 2. (**A**) Supernatants from the foot tissue homogenates were analyzed for pro-caspase-1, caspase-1(p10), pro-IL-1β, and IL-1β by immunoblotting. # indicates an individual sample. (**B**) The foot tissue homogenates were analyzed for IL-1β by ELISA. The values represent the means ± SD (*n* = 6 mice/group). *#*, significantly different from vehicle group, *p* < 0.05; *, significantly different from MSU alone, *p* < 0.05. (**C**) Representative pictures of immunohistochemistry staining of foot tissues for caspase-1 and IL-1β (200×).

**Figure 4 molecules-24-02138-f004:**
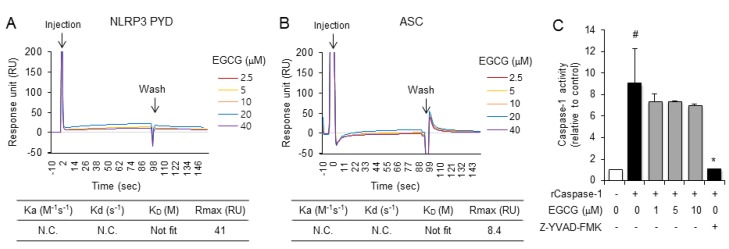
EGCG does not directly associate with NLRP3 and apoptosis-associated speck-like protein containing a CARD (ASC) or inhibit caspase-1 enzyme activity. (**A**,**B**) Sensorgrams were obtained by SPR analysis to determine the interaction between EGCG and recombinant (**A**) NLRP3-PYD or (**B**) ASC. Different concentrations of EGCG are presented as an overlay plot aligned at the start of injection. Table of kinetic parameters obtained from the SPR analysis and calculated by Biocore T200 evaluation software. (**C**) An *in vitro* assay for caspase-1 enzymatic activity was performed using a fluorometric caspase-1 assay kit with recombinant caspase-1 (rCaspase-1) in the presence or absence of EGCG or Z-YVAD-FMK. The values represent the means ± SD (*n* = 3). *#*, significantly different from vehicle, *p* < 0.05; *, significantly different from rCaspase-1 alone, *p* < 0.05.

**Figure 5 molecules-24-02138-f005:**
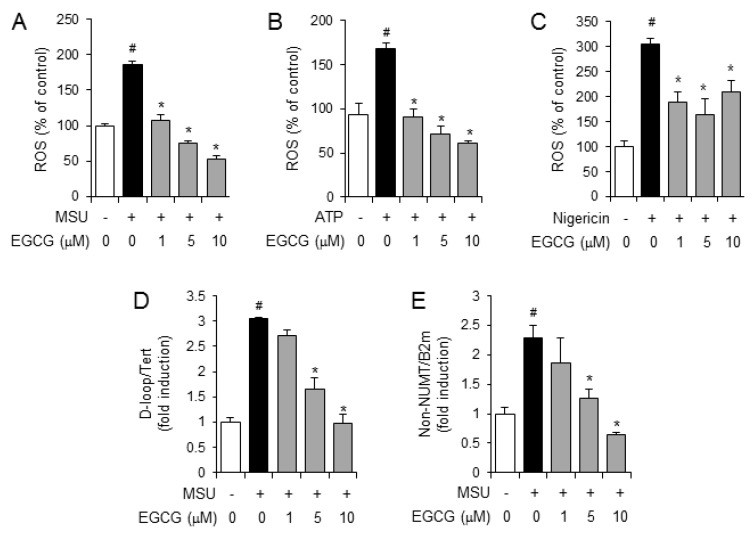
EGCG suppresses the production of reactive oxygen species and the *de novo* synthesis of mitochondrial DNA induced by NLRP3 inflammasome activators in primary macrophages. (**A**–**C**) Bone marrow-derived primary mouse macrophages were pretreated with an oxidative stress indicator, H_2_DCFDA (10 μM), for 1 h and further stimulated with monosodium uric acid (MSU) crystals (500 μg/mL), ATP (5 mM), or nigericin (10 μM) for 1 h in the presence or absence of EGCG. Reactive oxygen species (ROS) levels are presented as a percentage of the control. (**D**,**E**) LPS-primed bone marrow-derived primary mouse macrophages were stimulated with MSU crystals (500 μg/mL) for 6 h in the absence or presence of EGCG. The levels of mitochondrial DNA were determined by quantitative PCR. The values represent the means ± SD (*n* = 3); *#*, significantly different from vehicle group, *p* < 0.05; *, significantly different from MSU, ATP, or nigericin alone, *p* < 0.05.

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
