# Peer review of "Epigallocatechin-3-Gallate Prevents Acute Gout by Suppressing NLRP3 Inflammasome Activation and Mitochondrial DNA Synthesis"

_molecules, 2019, doi:10.3390/molecules24112138_

Round 1

Reviewer 1 Report

In this manuscript, Lee and co-authors show that epigallocatechin-3-gallate (EGCG) inhibits NLRP3 inflammasome activated by monosodium urate (MSU) crystals in mouse macrophages and in vivo. EGCG suppressed de novo synthesis of mitochondrial DNA, which is critical for the activation of NLRP3 inflammasomes. Consistently, the production of ROS was suppressed by the treatment with EGCG.

Overall, this manuscript is novel and interesting by discovering that EGCG is active in suppressing inflammasome induced by MSU by affecting mitochondrial DNA synthesis. The manuscript is concise and well written.

There are only several points that need to be addressed or discussed.

Specific comments.

1.        There are several papers that have previously shown that EGCG is potent to suppress NLRP3 inflammasome activation (Tsai et al. Free Radical Biology and Medicine 2011, Gao et al., PLoS One 2016). The authors are better to cite the papers in the introduction section.

2.        The authors show that the MSU-mediated cleavage of IL-1b was inhibited by EGCG treatment. It looks that high Pro-IL-1b expression is observed in Figure 3A. Does this mean that the authors primed the mice prior to the treatment with MSU, or Pro-IL-1b is produced even in resting foot tissues?

3.        The effect of EGCG in the synthesis of mitochondrial DNA is intriguing, though it is not clear if this effect is specific to MSU or applicable to other NLRP3 activators. The authors might examine the effect of EGCG on mitochondrial DNA synthesis in response to ATP or nigericin to clarify the point.

4.        The authors can discuss the mechanisms how EGCG controls de novo mitochondrial DNA synthesis to activate the inflammasome.

Author Response

We would like to express our sincere appreciation to the reviewer’s comments to improve our manuscript. We have tried our best to resolve each of the comments and hope the revision is satisfactory. We provide our point-to-point responses here and the changes made are shown in red-colored font in the revised manuscript.

1) Comment 1: There are several papers that have previously shown that EGCG is potent to suppress NLRP3 inflammasome activation (Tsai et al. Free Radical Biology and Medicine 2011, Gao et al., PLoS One 2016). The authors are better to cite the papers in the introduction section.

Response 1: As the reviewer suggested, we added these previous reports in Introduction section as follows:

There is a previous report that EGCG reduces the mRNA and protein expression of renal NLRP3 in a lupus nephritis mouse model [10]. Similarly, Gao et al. reports that EGCG decreases the protein level of renal NLRP3 in a contrast-induced nephropathy rat model [11]. These reports show that oral or intravenous infusion of EGCG suppresses the expression of tissue NLRP3 protein, possibly leading to the reduction of NLRP3 inflammasome activation. However, it is not entirely elucidated whether EGCG affects the activation of NLRP3 inflammasome induced by various NLRP3 inducers in macrophages and in a gouty inflammatory response.

2) Comment 2: The authors show that the MSU-mediated cleavage of IL-1b was inhibited by EGCG treatment. It looks that high Pro-IL-1b expression is observed in Figure 3A. Does this mean that the authors primed the mice prior to the treatment with MSU, or Pro-IL-1b is produced even in resting foot tissues?

Response 2: The animals were not treated with any specific priming reagent such as LPS. The foot tissues were not completely resting state since the footpad of the right hindfoot was subcutaneously injected with PBS (0.1 ml/mouse) as a vehicle or MSU crystals (2 mg/0.1 ml of PBS/mouse) to induce NLRP3 inflammasome activation. The footpad tissues were collected after 24 hr for further analysis. Since the animals were not restrained after subcutaneous injection to footpad, physical and chemical insult by needle and solution containing PBS or MSU might play a role in inducing pro-IL-1b production. Nonetheless, the basal pro-IL-1b induction itself does not affect pathological indicators such as foot thickness gain, histological assessment, and myeloperoxidase activity since control group (PBS+vehicle) did not show any pathological changes despite of basal pro-IL-1b production in immunioblot.

3) Comment 3: The effect of EGCG in the synthesis of mitochondrial DNA is intriguing, though it is not clear if this effect is specific to MSU or applicable to other NLRP3 activators. The authors might examine the effect of EGCG on mitochondrial DNA synthesis in response to ATP or nigericin to clarify the point.

Response 3: Zhong et al. reports that blockade of mitochondrial DNA (mtDNA) synthesis inhibited IL-1b production induced by NLRP3 activators such as ATP, nigericin, and MSU, while IL-1b production by an AIM2 inflammasome activator, poly dA:dT, was not inhibited by blockade of mtDNA synthesis [13]. These show that newly synthesized mtDNA is required for NLRP3 inflammasome activation induced by various NLRP3 activators [13]. Therefore, it is likely that the common inhibitory effect of EGCG on various NLRP3 inducers may be linked to the common suppression of mtDNA synthesis in response to various NLRP3 inducers including ATP or nigericin. This needs to be further elucidated in a future study.

We added this point in the Discussion.

4) Comment 4: The authors can discuss the mechanisms how EGCG controls de novo mitochondrial DNA synthesis to activate the inflammasome.

Response 4: Cytidine/uridine monophosphate kinase 2 (CMPK2) is an essential enzyme that provides deoxyribonucleotides for mtDNA synthesis. CMPK2 expression is regulated by IRF1 transcription factor activated in an LPS-priming step. CMPK2-dependent mtDNA synthesis is required for the production of oxidized mtDNA fragments in response to NLRP3 activators. Oxidized mtDNA associates with the NLRP3 inflammasome complex to activate NLRP3 inflammasome. We show that de novo synthesis of mtDNAs induced by MSU crystals is decreased by EGCG in macrophages. This suggests that EGCG may regulate the upstream events to produce mtDNA including CMPK2 activity or CMPK2 expression. It needs to be further pursued to investigate the detailed mechanism for inhibition of de novo synthesis of mtDNA by EGCG.

We added this point in the Discussion.

Reviewer 2 Report

In the manuscript entitled “Epigallocatechin-3-gallate prevents acute gout by suppressing NLRP3 inflammasome activation and mitochondrial DNA synthesis” the authors aim to show the beneficial effect of this compound in preventing acute gout by inhibiting NLRP3 inflammasome activation. The manuscript is very clear and well written and the sections are well organized to reach the point, but some minor revisions are necessary to enhance the quality.

Minor corrections

Abbreviations should be all written in full at first mention, in text and figure caption, this includes H&E, IL-1β, DMSO, LPS etc.

Authors are citing Figure 4. D and E in text (Line 183), while this is not represented in the actual figure 4

Line 44-45, a reference is required

Figure 5 caption can be modified to illustrate that “… EGCG suppresses production of reactive oxygen species (ROS) and mitochondrial DNA synthesis…”

Sources for all reagents, including H2DCFDA should be provided

In conclusion (Line 254-257), authors should also mention something on the inhibitory effect of ROS by EGCG

Author Response

We would like to express our sincere appreciation to the reviewer’s comments to improve our manuscript. We have tried our best to resolve each of the comments and hope the revision is satisfactory. We provide our point-to-point responses here and the changes made are shown in red-colored font in the revised manuscript.

1) Comment 1: Abbreviations should be all written in full at first mention, in text and figure caption, this includes H&E, IL-1β, DMSO, LPS etc.

Response 1:  As the reviewer suggested, abbreviations are written in full at first mention as follows:

-In line 44: interleukin (IL)-1b

-In line 81: lipopolysaccharide (LPS)

-In line 145: dimethyl sulfoxide (DMSO)

-In line 149: hematoxylin and eosin

-In line 170: surface plasmon resonance (SPR)

2) Comment 2: Authors are citing Figure 4. D and E in text (Line 183), while this is not represented in the actual figure 4.

Response 2: We apologize for the mistake. We corrected Figure 4D and E to Figure 5D and 5E.

3) Comment 3: Line 44-45, a reference is required.

Response 3: We have inserted a reference [3] in line 44.

4) Comment 4:  Figure 5 caption can be modified to illustrate that “… EGCG suppresses production of reactive oxygen species (ROS) and mitochondrial DNA synthesis…”

Response 4: We modified a Figure 5 caption as follows:

EGCG suppresses the production of reactive oxygen species and the de novo synthesis of mitochondrial DNA induced by NLRP3 inflammasome activators in primary macrophages.

5) Comment 5: Sources for all reagents, including H2DCFDA should be provided

Response 5: We added the source of H2DCFDA and other regents in “4.2 Reagents” as follows:

a) Nigericin and H2DCFDA were purchased from Sigma-Aldrich.

b) All other reagents, if not specified, were purchased from Sigma-Aldrich.

6) Comment 6: In conclusion (Line 254-257), authors should also mention something on the inhibitory effect of ROS by EGCG.

Response 6: As the reviewer suggested, we added the inhibitory effect of ROS by EGCG in conclusion as follows:

In conclusion, EGCG inhibits acute gout inflammation, including proinflammatory cytokine release and neutrophil infiltration into the lesion site, mediated by the suppression of NLRP3 inflammasome activation in macrophages. The inhibition of ROS production and mtDNA synthesis by EGCG contributes to its suppressive effect on the NLRP3 inflammasome. 

Reviewer 3 Report

General comments:

The author study the impacts of EGCG on the gout inflammation induced by three inducers. This is an informative study. However, some results need to be added in the manuscript and the results are contradicted.

Specific comments:

1.      Add house keeping protein results in Figure 1. Add the ratio of target proteins to the house keeping gene in Figure 1, otherwise, it is difficult to identify which one is upregulated and which one is inhibited, especially for E-LYS.

2.      According to Figure 1, MSU (A) inhibit pro-caspase-1 and Pro-IL-1b and EGCG upregulated them in LYS, but ATP (C) upregulated them and EGCG inhibited them. The results are contradicted. Please explain the reason.

3.      For the experiment of Figure 1, did you do the treatment that only treat macrophage with EGCG without any inducers?

4.      Figure 3 caption, “#, p<0.05 vs. PBS+vehicle; , p < 0.05 vs. MSU+vehicle” This is not clear. Please clarify what did you compare.

5.      Figure 3, what are other bands in (A)? Which one is the housing keeping protein band? If caspase-1(p10) and cleaved IL-1b decreased, should pro-caspase-1 and pro-IL-1b increase compared with MSU treatment?

6.      L286-287, LPS was used to treat BMDMs before MSU, ATP, and nigericin stimulation. Why the dose for MSU (500ng/ml) treatment is different from ATP and nigericin (100ng/ml) treatment?

Author Response

We would like to express our sincere appreciation to the reviewer’s comments to improve our manuscript. We have tried our best to resolve each of the comments and hope the revision is satisfactory. We provide our point-to-point responses here and the changes made are shown in red-colored font in the revised manuscript.

1) Comment 1: Add house keeping protein results in Figure 1. Add the ratio of target proteins to the house keeping gene in Figure 1, otherwise, it is difficult to identify which one is upregulated and which one is inhibited, especially for E-LYS.

Response 1: The cleavage degree of pro-caspase-1 and pro-IL-1b to caspase-1 and IL-1b, respectively, is the hallmark of inflammasome activation. Most of NLRP3 inflammasome activators do not affect the expression of pro-caspase-1 and pro-IL-1b, but induce the production of caspase-1 and IL-1b. Therefore, the comparison for the change of pro-form to cleaved form represents the degree of NLRP3 inflammasome activation.

To clarify upregulation or downregulation of NLRP3 inflammasome, we added the density graphs showing the ratio of caspase-1(p10) and cleaved IL-1b to pro-caspase-1 and pro-IL-1b, respectively, in Figure 1. The following description is added to Figure 1 legend:

In the bar graph, the band density of caspase-1(p10) and cleaved IL-1b normalized to corresponding pro-caspase-1 and pro-IL-1b, was expressed as means ± SEM (n=3).

2) Comment 2: According to Figure 1, MSU (A) inhibit pro-caspase-1 and Pro-IL-1b and EGCG upregulated them in LYS, but ATP (C) upregulated them and EGCG inhibited them. The results are contradicted. Please explain the reason.

Response 2: As mentioned in Response 1, most of NLRP3 inflammasome activators do not affect the expression of pro-caspase-1 and pro-IL-1b, but induce the production of caspase-1 and IL-1b. The difference in pro-caspase-1 and Pro-IL-1b seems to be an experimental variation. Since the activation of NLRP3 inflammasome is determined by the comparison of the change of pro-form to cleaved form, we added bar graphs comparing the ratio of caspase-1(p10) and cleaved IL-1b to pro-caspase-1 and pro-IL-1b, respectively, to provide the activation and inhibition of NLRP3 inflammasome more clearly.

3) Comment 3: For the experiment of Figure 1, did you do the treatment that only treat macrophage with EGCG without any inducers?

Response 3: The details on the treatment are described in “4.3 Analysis of inflammasome activation” and Figure 1 legend. Macrophages were treated with EGCG for 1 hr after washing out LPS. Then, the cells were stimulated with NLRP3 inducers in the presence or absence of EGCG as indicated in Figure 1.

To make it clearer, we added “in the presence or absence of EGCG” in Figure 1 legend and Methods section as follows:

a) In line 102-105: LPS-primed bone marrow-derived primary macrophages were treated with EGCG for 1 hr and then stimulated with monosodium uric acid (MSU) crystals (500 μg/ml for 6 hr), ATP (5 mM for 1 hr), or nigericin (10 mM; 2 hr for E and 16 hr for F) in the presence or absence of EGCG as indicated.

b) In line 317-319: The cells were treated with EGCG for 1 hr and stimulated with NLRP3 inflammasome activators such as MSU crystals, ATP, and nigericin in the presence or absence of EGCG in serum-free medium.

4) Comment 4: Figure 3 caption, “#, p<0.05 vs. PBS+vehicle; , p < 0.05 vs. MSU+vehicle” This is not clear. Please clarify what did you compare.

Response 4: As the reviewer suggested, we revised the Figure 3 caption as follows:

#, significantly different from vehicle group, p <0.05. *, significantly different from MSU alone, p <0.05.

In addition, we revised the other figure captions in a similar way.

5) Comment 5: Figure 3, what are other bands in (A)? Which one is the housing keeping protein band? If caspase-1(p10) and cleaved IL-1b decreased, should pro-caspase-1 and pro-IL-1b increase compared with MSU treatment?

Response 5: Firstly, other bands in (A) are non-specific proteins. We identified pro-caspase-1, pro-IL-1b, caspase-1(p10), and cleaved IL-1b with the corresponding molecular weight.  

Secondly, as described in Response 1, the comparison for the change of pro-form to cleaved form represents the degree of NLRP3 inflammasome activation.

Thirdly, not all pro-caspase-1 and pro-IL-1b proteins degrade into caspase-1(p10) and cleaved IL-1b upon NLRP3 activation to exert the consequent biological activity. We do not usually observe a prominent change in pro-caspase-1 and pro-IL-1b in response to NLRP3 inducers in macrophages, in consistent with other published reports.

6) Comment 6: L286-287, LPS was used to treat BMDMs before MSU, ATP, and nigericin stimulation. Why the dose for MSU (500 ng/ml) treatment is different from ATP and nigericin (100ng/ml) treatment?

Response 6: LPS is added to prime macrophages, inducing the expression of NLRP3 and pro-IL-1b. Different types of NLRP3 inducers require different levels of LPS priming. We have selected the optimized dosage of LPS for each NLRP3 inducer through the accumulated experimental experience.

Round 2

Reviewer 3 Report

1.       For western blotting experiments, if you don’t show housing keeping protein band, how can others know whether you loaded same amount of protein for all the treatments?

2.       Add statistic notation in bar graph of Figure 1.

3.       L113, SEM is a measure of precision for an estimated population mean and SD is a measure of data variability around mean of a sample of population. What you want to present here is the dispersion of your replicate to your sample means.  It is not correct to use SEM instead of SD. Please change all your data presentation from mean+/-SEM to SD.

Author Response

1) Comment 1: For western blotting experiments, if you don’t show housing keeping protein band, how can others know whether you loaded same amount of protein for all the treatments?

Response 1: We understand the reviewer’s concern. In most inflammasome experiments, the levels of pro-caspase-1 and pro-IL-1b could serve as loading controls since the protein amounts of pro-caspase-1 and pro-IL-1b are not much affected by NLRP3 inflammasome activators such as ATP, MSU, and nigericin. Therefore, the immunoblots of pro-caspase-1 and pro-IL-1b should be presented together with the immunoblots of cleaved caspase-1 and IL-1b.

2) Comment 2: Add statistic notation in bar graph of Figure 1.

Response 2: We added statistic notation in bar graph of Figure 1A, 1C, and 1E and revised Figure 1 legend accordingly.

3) Comment 3: L113, SEM is a measure of precision for an estimated population mean and SD is a measure of data variability around mean of a sample of population. What you want to present here is the dispersion of your replicate to your sample means. It is not correct to use SEM instead of SD. Please change all your data presentation from mean+/-SEM to SD.

Response 3: As the reviewer suggested, all of the figures are changed to mean+/-SD. Figure legends and Methods section are revised accordingly.